# Leprosy as immune reconstitution inflammatory syndrome in patients living with HIV: Description of French Guiana's cases over 20 years and systematic review of the literature

**Alice Mouchard**[1]*, **Romain Blaizot**[2], **Jenna Graille**[1], **Pierre Couppié**[1,2], **Chloé Bertin**[1]

**1** Service de Dermatologie, Centre hospitalier de Cayenne, Cayenne, French Guiana, **2** Tropical Biome and Immunophysiopathology (TBIP), Université de Lille, CNRS, INSERM, Institut Pasteur de Lille, U1019-UMR9017-CIIL-Centre d'Infection et d'Immunité de Lille, Centre Hospitalier de Cayenne, Université de Guyane, Cayenne, French Guiana

* aae.mouchard@gmail.com

**Data Availability Statement:** All relevant data are within the manuscript and its Supporting Information files.

## Abstract

### Background

HIV infection is highly prevalent in French Guiana, a territory where leprosy is also endemic. Since the introduction of Highly Active Antiretroviral Treatment (HAART) in the management of HIV, leprosy has been reported as part of the immune reconstitution inflammatory syndrome (IRIS).

### Methodology/Principal findings

We aimed to present a general description of these forms of leprosy as IRIS, highlighting clinical and therapeutic specificities. A retrospective study was conducted in French Guiana, including patients living with HIV (PLHIV) with advanced infection (CD4 < 200/mm3) and developing leprosy or a leprosy reaction within six months of HAART initiation, from 2000 to 2020. Clinical, histological and biological data were collected for all these patients. Six patients were reported in French Guiana. A systematic review of the literature was conducted, and its results were added to an overall analysis. Overall, seventy-three PLHIV were included. They were mainly men (74%), aged 22–54 years (median 36 years), mainly from Brazil (46.5%) and India (32.8%). Most leprosy cases (56.2%) were borderline tuberculoid (BT). Leprosy reactions were frequent (74%), mainly type 1 reaction (T1R) (68.5%), sometimes intense with ulceration of skin lesions (22%). Neuritis was observed in 30.1% of patients. The outcome was always favorable under multidrug therapy (MDT), continuation of HAART and additional corticosteroid therapy in case of neuritis or ulceration. There was no relapse.

### Conclusion

Leprosy as IRIS in PLHIV mainly presents as a BT leprosy in a T1R state, sometimes with ulcerated skin lesions. Response to MDT is usually good. Systemic corticosteroids are necessary and efficient in case of neuritis.

**Funding:** The author(s) received no specific funding to this work.

**Competing interests:** The authors have declared that no competing interests exist.

## Author summary

Leprosy is an infection caused by *Mycobacterium leprae* characterized by skin and nerve lesions. Leprosy reactions can be observed, depending on variations in host-specific cellular immunity. Leprosy is described after antiretroviral therapy (HAART) initiation in immunocompromised PLHIV from countries where leprosy and HIV infections are endemic. This is known as immune restoration inflammatory syndrome (IRIS), a brutal inflammatory response directed against a latent or quiescent pathogen. In this study we searched for cases of leprosy as IRIS in French Guiana and those published in the literature in order to describe their clinical characteristics. Overall, our results show that these cases are mostly observed as borderline tuberculoid leprosy, associated with or quickly followed by a type 1 leprosy reaction, sometimes with neuritis and/or ulceration of lesions. The outcome is favorable under standard leprosy treatment with HAART maintenance.

## Introduction

Leprosy is a chronic infection that is far from being eliminated with more than 200,000 new cases per year reported worldwide in 2019. It remains a major public health problem in terms of physical and social disability, particularly in South America, South East Asia and Africa [1]. In many of these countries where leprosy is endemic, HIV infection is highly prevalent. However, there are few epidemiological data on leprosy-HIV co-infection. Studies conducted in the early-to-mid-1990s suggested that co-infection with HIV did not alter the incidence and clinical spectrum of leprosy and that each disease progressed independently [2]. Since the introduction of Highly active antiretroviral treatment (HAART) in HIV management, leprosy has been reported as part of the immune reconstitution inflammatory syndrome (IRIS) in areas where the two diseases overlap. The first case reported in 2003 [3]. IRIS is an inappropriate inflammatory response to an infection that occurs in severely immunocompromised PLHIV (CD4 cell count $< 200/mm^3$) within the first six months of HAART initiation. In 2008, Deps and Lockwood [4] proposed a definition of leprosy as an IRIS to facilitate its identification and recognition: (1) leprosy and/or leprosy reaction presenting within the six months of starting HAART; (2) advanced HIV infection; (3) low CD4+ T lymphocyte (CD4) count before starting HAART; (4) CD4 count increase after HAART initiation. In 2020, among 37.7 million PLHIV across the world, 73% had access to antiretroviral treatment [5]. With the widespread availability of HAART worldwide, it was expected that leprosy as an IRIS would be increasingly reported. However, this assumption has yet to be confirmed.

French Guiana is a South American territory of almost 300,000 inhabitants of diverse origins were leprosy is a re-emerging public health problem, with a prevalence rate of 1 / 10,000 inhabitants from 2007 to 2014 [6]. HIV prevalence is also high, affecting about 1% of the population [7]. The occurrence of leprosy as IRIS has not been studied in this territory.

Our aim was to provide a general description of leprosy as IRIS among PLHIV, highlighting their clinical features and treatment modalities. In the following article, we first report all cases of IRIS leprosy that occurred in PLHIV in French Guiana between 2000 and 2020. As the most recent review of the world literature on the subject was conducted a decade ago by Deps and Lockwood [8], we also present an updated systematic review of the literature.

## Materials and methods

### Retrospective study in French Guiana

The Dermatology Department of the Cayenne Hospital Center (Andrée Rosemon) is the only structure for diagnosis and monitoring of patients with leprosy in French Guiana. Patients are followed up in Cayenne, or during missions conducted by the dermatology team in remote health centers (Saint-Laurent du Maroni, Saint-Georges de l'Oyapock, Maripasoula). These on-field missions allow us to manage patients living in the rainforest hinterland of French Guiana. We searched the files of all patients followed for leprosy in the Dermatology Department and extracted all files of HIV infected patients meeting the criteria defined by Deps and Lockwood[4]. Patients with leprosy and/or leprosy reactions occurring before HAART initiation or more than six months later were excluded. Collected data included CD4 cell count, HIV viral load (before and after HAART), history of opportunistic infections, country of birth. Leprosy and leprosy reaction treatment were collected as well as efficacy and tolerance during follow-up.

Diagnosis of leprosy was based on clinical signs and histopathology. Leprosy forms were classified according to the clinical and pathological Ridley-Jopling scale [8] and classified as paucibacillary (PB) or multibacillary (MB) according to the WHO classification based on the number of lesions and the presence or absence of Acid Fast Bacilli(AFB) at skin smear examination [9].Leprosy as IRIS was classified according to Deps and Lockwood's classification based on the time when the phenomenon takes place regarding HAART and MDT initiations [8].

Data were analyzed with EXCEL. This project was authorized under the CNIL registration number 2215827 and fully complied with the French legal ethics requirements.

### Systematic review: Data sources, search strategy and selection process

We conducted a search of PubMed and EMBASE to identify all published cases of IRIS as leprosy with no time limit. The search had no language restrictions. The key search terms used were "leprosy" and "HIV/AIDS" or "immune reconstitution inflammatory syndrome" using Medical Subject Headings (MeSH). Bibliographic references of selected articles and grey literature (Google Scholar) were used for non-indexed journal articles. This systematic review has been registered in the international prospective register of systematic review (PROSPERO) under the registration number: CRD4 CRD42021267703. The cases that met the definition of Deps and Lockwood mentioned above were selected. The absence of CD4 cell data or a CD4 cell count > 200 cells/mm$^3$ was not an exclusion criterion if clinical and chronological data were met. Nevertheless, we did not include patients when the delay between HAART initiation and the occurrence of IRIS was not mentioned or when clinical information was insufficient. Collected data included CD4 cell count, HIV viral load (before and after HAART), history of opportunistic infections, country of birth. Two researchers (AM, CB) independently screened, retrieved and analyzed each report. In case of disagreement, consensus was reached by discussion.

## Results

### Retrospective study in French Guiana

**Patients characteristics and treatment.** From January 1$^{st}$ 2000 to June 1$^{st}$2020 246 patients were followed for leprosy in French Guiana. Twenty-two were PLHIV and six met the criteria of leprosy as IRIS (Fig 1). Six male patients aged 24 to 54 (mean 42 years) were included (Table 1). One patient was born in French Guiana, two were from Haiti and three

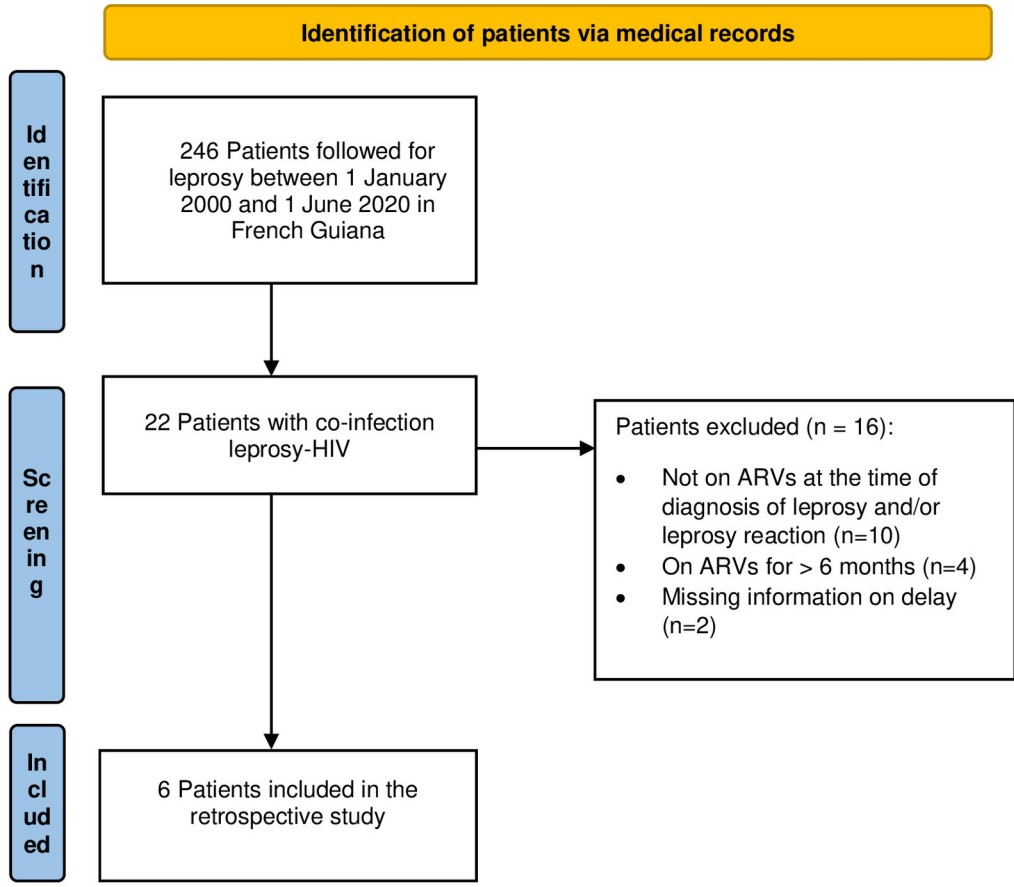

**Fig 1. Flow chart of the retrospective study of leprosy as IRIS in French Guiana between 2000 and 2020.**

**Table 1. Epidemiological, clinical and biological characteristics of the 6 patients included in the retrospective study diagnosed with leprosy as IRIS in French Guiana between 2000 and 2020.** M: male; RNA: ribonucleic acid; SL: skin lesion(s); T1R: type 1 reaction; U: ulceration; NT: neuritis; NFI: nerve fonction impairment; Se: sensitive; Mo: motor; [1]: before MDT introduction; [2]: after MDT introduction; TT: polar tuberculoid; BT: borderline tuberculoid; BB: borderline borderline; PB: paucibacillary; MB: multibacillary.

| Patient | Origin | Sex | Age | Delay between HAART and IRIS (weeks) | CD4 at initiation of HAART (cells/mm3) | CD4 at the onset of IRIS (cells/mm3) | Fold increase of CD4 | Plasma HIV RNA level at initiation of HAART (copies/ml) | Plasma HIV RNA level at the onset of IRIS (copies/ml) | Clinical manifestations | Ridley-Jopling | IRIS classification | MDT duration (months) | Additional treatment |
|---|---|---|---|---|---|---|---|---|---|---|---|---|---|---|
| 1 | Haïti | M | 54 | 6 | 87 | 257 | 3,0 | 19000 | 650 | SL/T1R$^1$/ U$^2$ | BB$^1$/BT$^2$ | 1 | 6 | No |
| 2 | French guiana | M | 40 | 14 | 130 | 278 | 2,1 | 40701 | 68 | SL/T1R$^1$/U$^1$/NT + NFI Se$^1$ | TT | 1 | 6 | Prednisone |
| 3 | Haiti | M | 44 | 24 | 105 | 268 | 2,6 | 159000 | <50 | SL | TT | 1 | 18 | No |
| 4 | Brazil | M | 24 | 1 | 28 | 50 | 1,8 | 297000 | 6000 | SL | BT | 1 | 12 | No |
| 5 | Brazil | M | 47 | 4 | 5 | 135 | 27,0 | 8300 | 1400 | SL/T1R$^2$/NT + NFI Se/Mo$^2$ | BT | 4 | 6* | Prednisone |
| 6 | Brazil | M | 46 | 10 | 25 | 96 | 3,8 | 59912 | <50 | SL/T1R$^1$/ U$^2$ | BT | 1 | 18 | No |

*: Patient 5 was lost to follow-up after this period of time

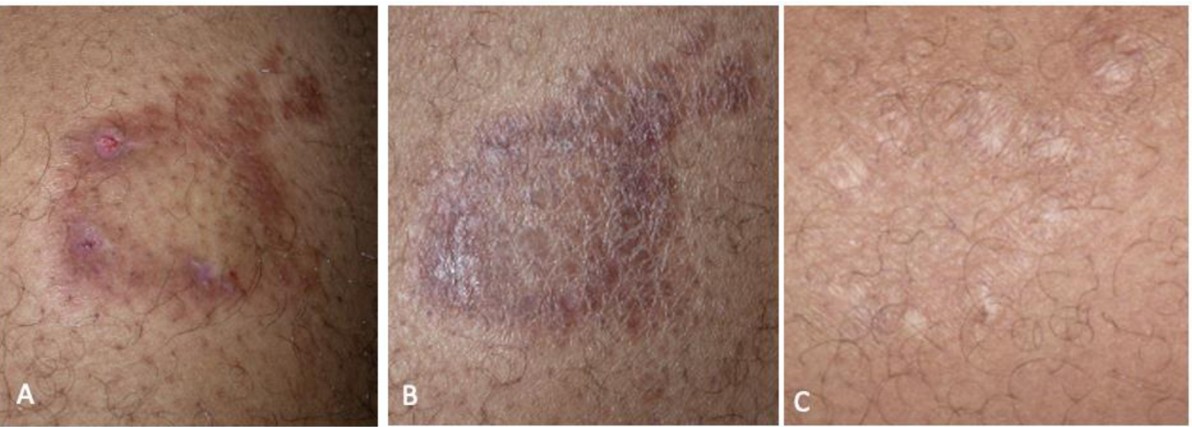

**Fig 2. IRIS type 1 after introduction of HAART in patient 3: TT on left flank.** A: at diagnosis of leprosy; B: on MDT and HAART; C: cured at M18 (continued MDT and HAART).

from Brazil. None of them had a history of leprosy before IRIS. Leprosy lesions were observed on average 10 weeks (1–24) after HAART initiation. All patients had a CD4 cell count < 200 / mm³ before HAART. Median CD4 cell count before HAART initiation and at the onset of IRIS were respectively 72 and 210/mm³. The median fold increase of CD4 before and after initiation of HAART was 2.8. The HIV viral load decreased by at least 1 log between HIV diagnosis and IRIS onset. The aspect of lesions at diagnosis was poly-morphic, with discrepancies between clinical and pathological findings in some patients (Figs 2–5). The mean duration of MDT was 11 months. There was no relapse recorded. We recorded no drug interactions between MDT and HAART and no adverse effects of corticosteroid therapy.

## Systematic review

The first published case of leprosy as IRIS after HAART initiation was a BT leprosy in a type 1 reaction state in a man from Uganda, reported in 2003 by Lawn *et al.*[3]. Since then, as HAART has become more readily available in countries were HIV and leprosy overlap, 70 cases of leprosy as IRIS have been published up to date in 40 publications [3,10–48] (Fig 6). Among the six cases described in our report, three were previously reported by our dermatol-ogy team: patients 1 and 2 by Couppié *et al.* in 2004 [10] and patient 3 by Sarazin *et al.* in 2005 [14]. In total 73 patients were analyzed.

## Patients characteristics (Tables 2 and S1)

Among the 73 patients analyzed, 54 (74%) were men with a mean age of 36 years, (range 22–54 years). Thirty-four patients (46.5%) were from Brazil, 24 (32.8%) from India, three (4.1%) from Haiti, two from Mexico (2.7%), two from Mali (2.7%) and the other seven patients were from the following countries: French Guiana, Venezuela, Angola, Senegal, Ivory Coast, Uganda, USA, and Philippines (1.4% for each).

At the time of leprosy diagnosis, 41 (56.2%) patients had a histopathological diagnosis of BT leprosy. Five patients initially presented with BL [25,32] or BB [10] leprosy, which then developed into BT leprosy after a few weeks of adding MDT to HAART. According to Deps et Lockwood [49], all these patients were classified as BT leprosy. Regarding the other patients: ten patients (13.7%) had a diagnosis of BL leprosy, eight (10.9%) of TT leprosy, seven (9.6%) of

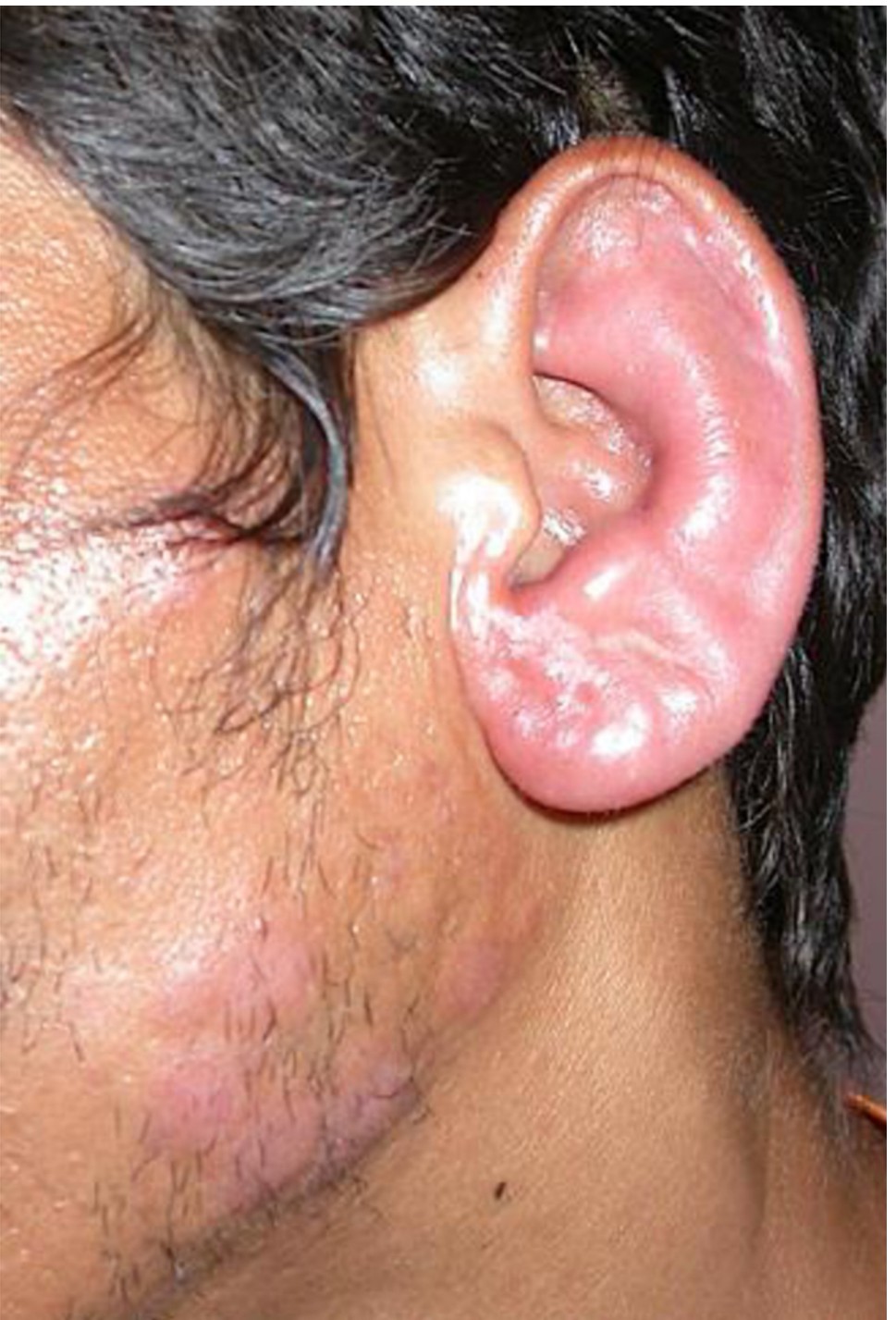

**Fig 3. IRIS type 1 after introduction of HAART in patient 4: BT leprosy on the left ear and cheek with hypertrophy of the great auricular nerve.**

BB leprosy, one (1.4%) of LL and one (1.4%) of neural leprosy. Data were not available for five patients (6.8%). According to the WHO classification, 39 (53.4%) patients had a PB leprosy and 34 (46.6%) had MB leprosy.

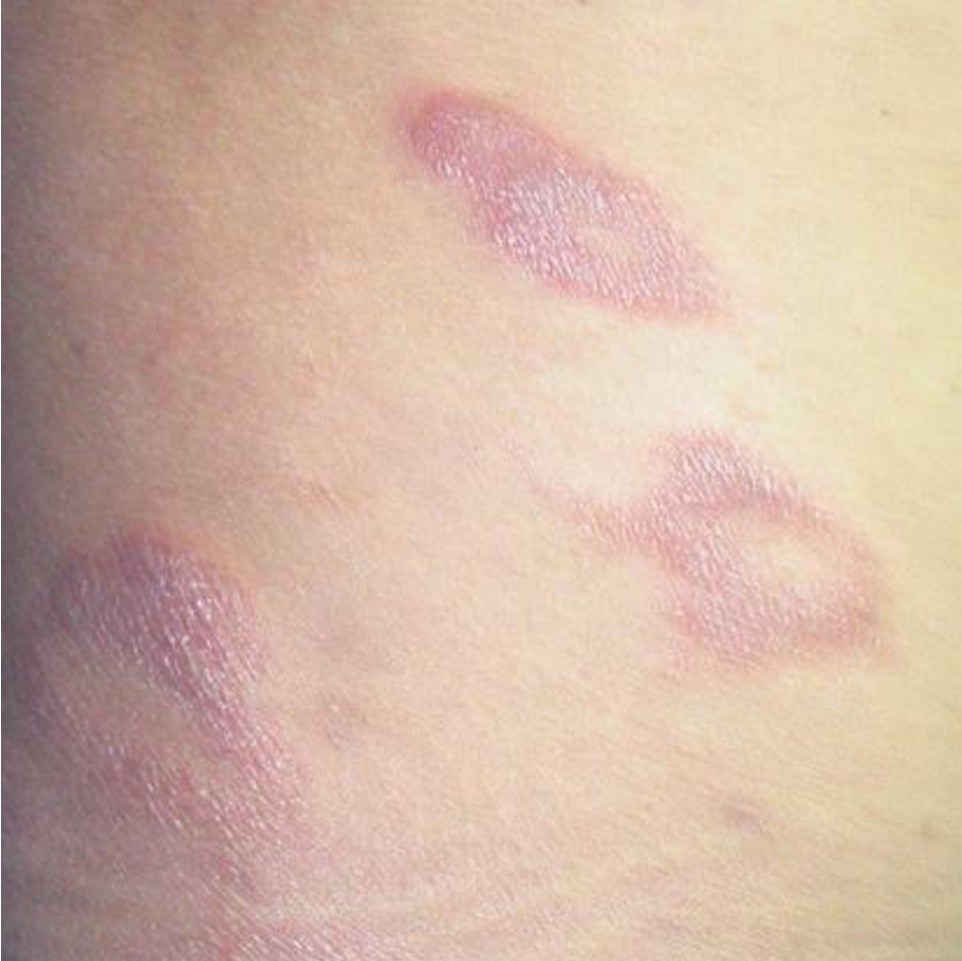

**Fig 4. IRIS type 4 after introduction of HAART in patient 5: BT leprosy in T1R on the right flank.**

Four patients had a clinical history suggestive of leprosy prior to HAART initiation, but the diagnosis was not made at that time. In these four patients, initial leprosy was diagnosed in the context of a T1R [12,23,28,42]. Two patients already had leprosy or a history of leprosy: one had BL leprosy with a T2R treated before HAART initiation, followed by the development of T1R on HAART [17]; another patient had BL leprosy treated several years earlier, and presented a BT leprosy in a reactive state at the time of HAART initiation [21]. The remaining 67 patients (92%) had no lesions suggestive of leprosy prior to HAART initiation. There was no mention of possible contact with leprosy index cases.

Skin lesions of leprosy in these PLHIV were typical except for two patients who presented diffuse eczematous papules hardly suggestive of leprosy [32]. Patient 6's lesions were suggestive of cutaneous leishmaniasis (Fig 5). For many patients, clinical distinction between the different forms of leprosy was difficult, and the diagnosis was possible only with histopathology.

The majority of patients presented leprosy reactions (54 patients, 74%). Among them 50 patients (68.5%) had T1R. The majority (36/50) of T1R occurred at the time of IRIS diagnosis, 6/50 occurred after MDT initiation and in 8/50 cases the time of onset was not specified. Four patients (5.5%) developed T2R: two cases occurred at the time of IRIS diagnosis and time of onset was not specified for the two others.

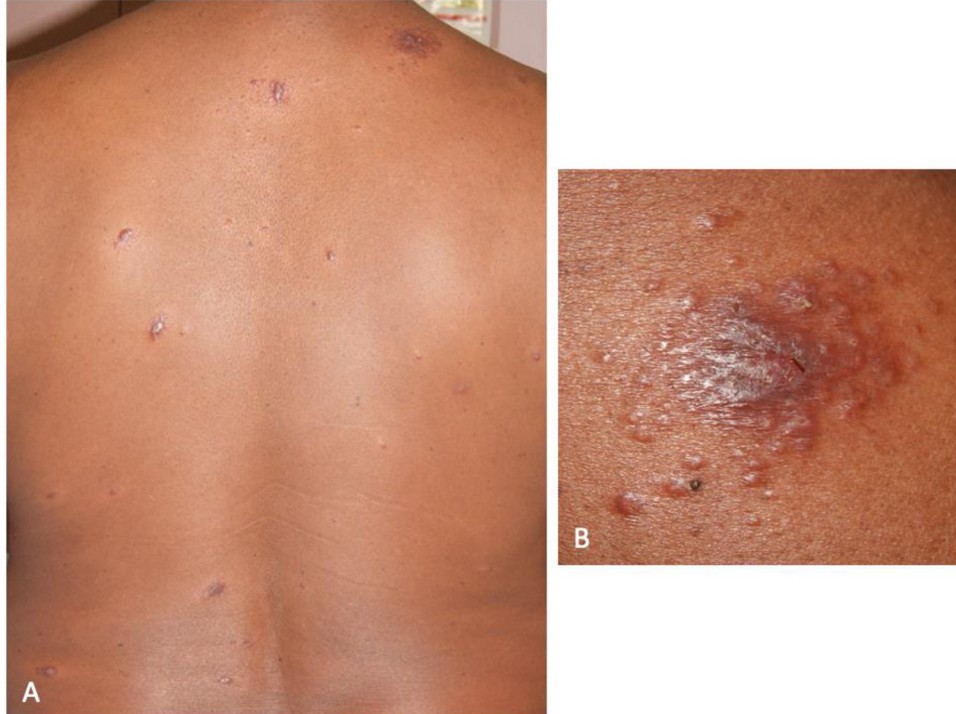

**Fig 5. IRIS type 1 after introduction of HAART in patient 6: BT leprosy in T1R on the trunk.** A: multiple inflammatory papulo-nodular lesions. B: close-up view of the lesion on the right shoulder.

A third of patients (22 patients, 30.1%) had neuritis (NT) associated with leprosy reaction. Among them 11 patients had sensitive and/or motor nerve function impairment (NFI). One patient presented a severe neuritis complicated with ulnar nerve abscess [44].

Sixteen patients (22%) had ulcerated/necrotic lesions. Pathological examinations were available for only three of them, showing a polymorphic inflammatory infiltrate and focal necrosis [12,19].

### Evidence of immune restoration and timing of onset

As described in our cases from French Guiana, immune restauration was constant after HAART initiation with a median fold increase of CD4 of 3.0. In one patient reported by Batista *et al*., immune reconstitution was histologically proven with increased granulomatous reaction and CD4-cell infiltrate before and after IRIS [23]. Mean time between HAART initiation and IRIS was 2.5 months (median 2 months, range 0.25–6) (Fig 7). IRIS associated with leprosy reaction (T1R or T2R) occurred 2 weeks earlier than non-reactive leprosy (8 weeks versus 10 weeks), probably as part of a stronger immune response reflected by a greater increase in CD4 count (median fold increase: +0.7).

According to Deps and Lockwood classification [49], most patients (46 patients or 67.1%) had IRIS type 1, i.e. an inflammatory syndrome unmasking previously untreated unknown leprosy. Two patients (2.7%) had IRIS type 2, two patients (2.7%) had IRIS type 3, and six patients (4.2%) had IRIS type 4. Information was not analyzable for 17 patients (23.4%) because of missing information on the prior presence of lesions suggestive of leprosy (Table 2).

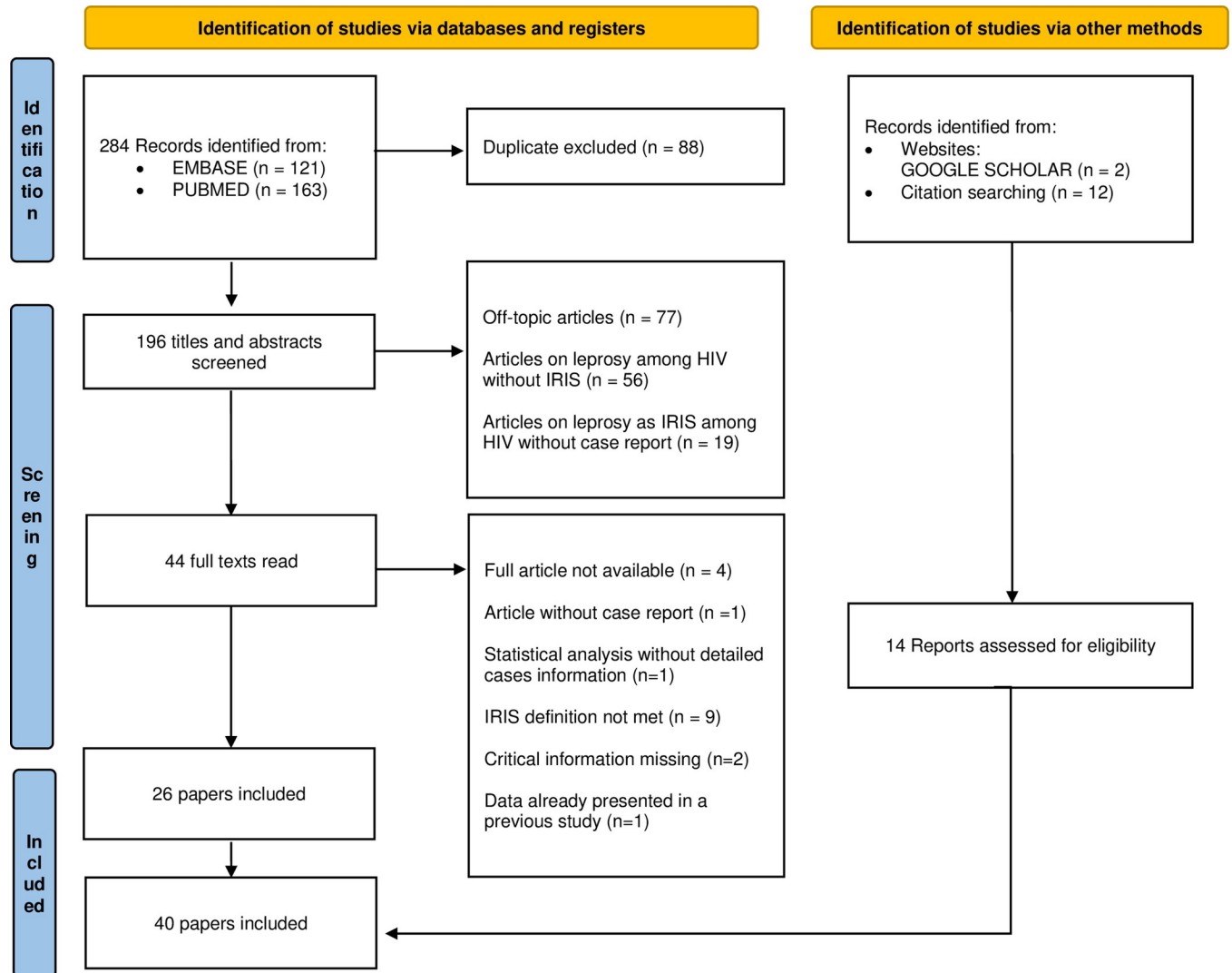

**Fig 6. Flow chart of the systematic review of leprosy as IRIS according to the PRISMA 2020 criteria.**

## Outcome

Due to the recurrent lack of information in the published cases on the modalities and duration of MDT treatment, we assumed that the authors were following the WHO recommendations in force at the time. MDT treatment was mentioned for 64 patients (87.7%). The evolution of leprosy lesions under MDT was favorable for all patients treated (Table 2). Thirty-three patients (45%) received with systemic corticosteroids, mainly because of leprosy reactions and neuritis, with a favorable outcome in most cases. Prolonged corticosteroid therapy was needed in several patients from the same paper [32]. In two cases, the addition of azathioprine [3] because of steroid dependence or thalidomide [24] was necessary, with a favorable outcome. Corticosteroids were efficient in treatment of neuritis, excepting one patient with long-lasting nerve damage of a foot levator [17]. Of the 16 patients who developed ulcerated lesions, four patients (including two in our case series) had spontaneous healing without the need for anti-inflammatory treatment. The other 12 were started on corticosteroid therapy as soon as ulcerations were noticed [10,12,20]. There were no reports of side effects of corticosteroids in

**Table 2. Number and proportion of patients diagnosed with leprosy IRIS included in the systematic review.**

|  | Number of patients, proportion (%) |
|---|---|
| HISTORY OF LEPROSY | |
| Yes (or probable) | 6 (8%) |
| No (or missing data) | 67 (92%) |
| RIDLEY-JOPLING CLASSIFICATION | |
| TT | 8 (10.9%) |
| BT | 41 (56.2%) |
| BB | 7 (9.6%) |
| BL | 10 (13.7%) |
| LL | 1 (1.4%) |
| Neural | 1 (1.4%) |
| Missing data | 5 (6.8%) |
| WHO CLASSIFICATION | |
| PB | 39 (53.4%) |
| MB | 34 (46.6%) |
| LEPROSY REACTIONS | |
| T1R | 50 (68.5%) |
| T2R | 4 (5.5%) |
| No | 19 (26%) |
| NEURITIS | |
| Yes | 22 (30.1%) |
| Doubt | 1 (1.4%) |
| No | 50 (68.5%) |
| ULCERATION | |
| Yes | 16 (22%) |
| No | 57 (78%) |
| IRIS CLASSIFICATION | |
| 1 | 46 (63%) |
| 2 | 2 (2.7%) |
| 3 | 2 (2.7%) |
| 4 | 6 (8.2%) |
| Missing data | 17 (23.4%) |
| FAVORABLE OUTCOME | |
| Yes | 64 (87.7%) |
| Missing data | 9 (12.3%) |

patients, apart from one case of staphylococcal sepsis following intravenous corticosteroid courses [32].

## Discussion

In PLHIV from countries where leprosy is endemic, initiation of HAART may unfold leprosy. These cases of leprosy will most often present as paucibacillary BT forms, associated with or or quickly followed by a T1R, sometimes with severe ulcerated skin lesions. Seventy-three cases of leprosy presenting as IRIS in PLHIV were described since the first case was published in 2003, including the six cases from our retrospective study in French Guiana. This number may seem low, but it is likely that many cases remain unpublished, and several cases reported were not relevant as they were occurring more than six months after HAART initiation [49]. In

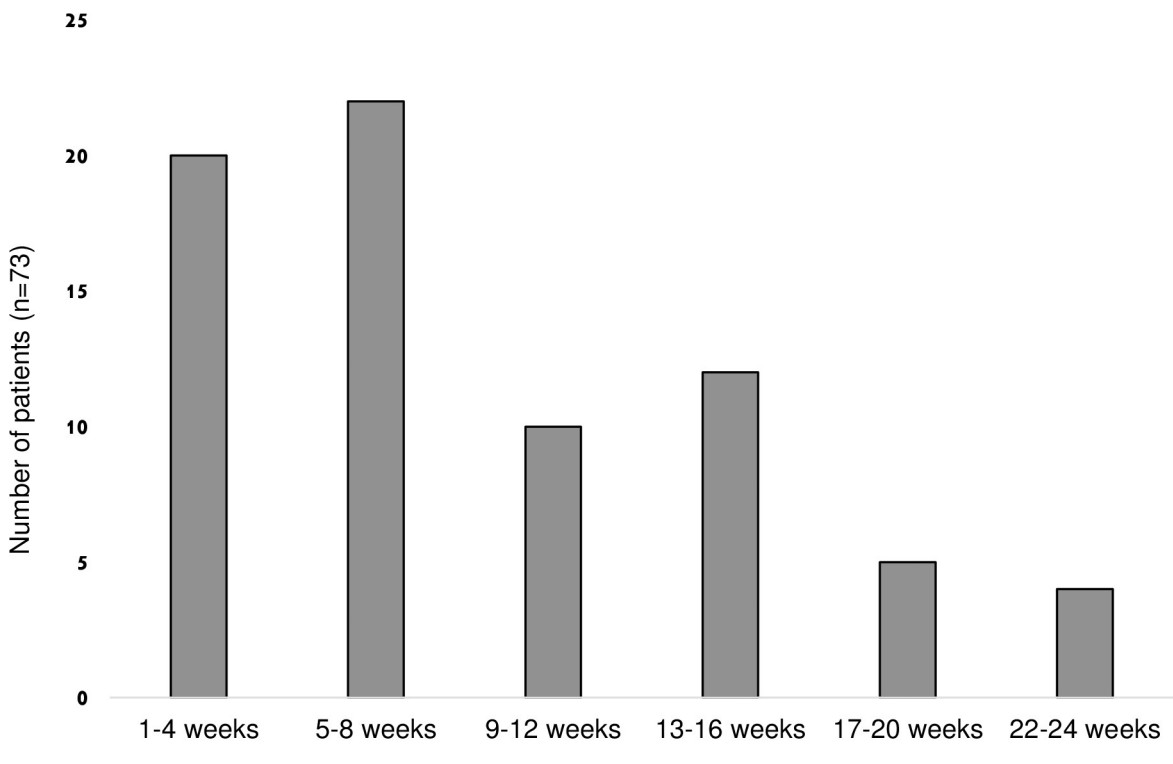

**Fig 7. Distribution of patients according to time from HAART introduction to leprosy IRIS.**

most patients (92%), HAART's initiation revealed previously unknown leprosy, mainly of the BT form (56.2%), associated with or quickly followed by a T1R (68.5%). These findings are in line with other reviews of PLHIV and leprosy [49,50]. The average time of 2.5 months for IRIS onset after HAART initiation is consistent with the data of Couppié *et al.*in French Guiana and with the global definition of IRIS [51].This delay in onset is close to the three months delay classically reported for IRIS associated with other mycobacteria, notably tuberculosis [52,53].

Most patients in the systematic review were reported in Brazil and India, two highly populated countries with high incident rates for leprosy and HIV. As described in the literature, most of the reported patients are young males (average age of 36 years), raising concerns about underdiagnosis of leprosy among women for socio-cultural reasons[1,32,50,54,55,56].

One possible explanation for the low incidence of leprosy in PLHIV is that *M.leprae* is susceptible to antibiotics used in the prevention or treatment of opportunistic infections: dapsone for pneumocystis and toxoplasmosis, rifampicin for tuberculosis, clarithromycin for *Mycobacterium avium complex* (MAC) infections [57,58]. On the other hand, one can argue that giving these antibiotics to patients with quiescent sub-clinical leprosy may lead to the release of *M. leprae* antigens and trigger a leprosy reaction [18]. The effects of drug interactions between HAART and MDT are probably limited by the fact that rifampicin, a potent inducer of cytochrome P450-314, is taken only once a month.

The predominance of BT and TT leprosy and the presence of T1R in two thirds of the cases suggest a strong cellular immune response. These findings are consistent with the increase in CD4 count and decrease in viral load under HAART. Most patients had IRIS type 1:

unmasking leprosy from a subclinical *M. leprae* infection. Since leprosy has a long incubation period, HAART, could act as an immunological trigger for a "premature" presentation of leprosy by improving both innate and cellular immune responses. This would explain the frequency of T1R due to an excess of pro-inflammatory response, predominantly Th1 [2].

Regarding leprosy reactions, the majority of the patients developed T1R, which is expected since borderline forms are immunologically unstable. Interestingly, one third of these patients with T1R had ulcerated lesions. No epidemiological data on this phenomenon in leprosy are available in the literature. Secondary ulceration is caused by a dermal edema following an excessive immune response against *M.leprae* [55,59]. De Oliveira *et al.* observed that during an episode of T1R, the number of activated CD8+ T lymphocytes (CD8) as well as perforin/granzyme B production were higher in the skin lesions of PLHIV on HAART co-infected patients than in those of mono-infected leprosy patients [60]. This could explain the intensity of inflammation in leprosy reactions associated with IRIS. IRIS is probably related to an early restoration of memory T-cell activity leading to an excessive inflammatory response in the presence of a latent infectious agent [61].

Systemic corticosteroid therapy in leprosy reactions is the treatment of choice to avoid neural damage [55,62,63]. According to WHO guidelines, ulceration classifies a leprosy reaction as severe, warranting the introduction of systemic corticosteroid therapy[62]. Nevertheless, spontaneous healing without corticosteroids was observed in four patients (4/16 or 25%), which could lead to propose therapeutic abstention for isolated ulcerations without neuritis. Xavier *et al.* observed a faster regression of neuritis under corticosteroids in PLHIV patients than in other leprosy cases [64]. Pires *et al.* found that PLHIV on HAART often had shorter leprosy reactions compared to non-HIV patients possibly due to improved cellular immunity [55].

Studies that have examined histologic and immunological parameters in small cohorts of leprosy-HIV co-infected patients have yielded conflicting results, making it impossible to draw definitive conclusions, although some authors have suggested an increase in the CD8 pathway [60,65,66,67,68]. It is important to note that cells present in peripheral blood do not necessarily reflect the number or function of these cells at the sites of infection, and that skin and/or nerve immunological investigations are essential. Infectious side effects from systemic corticosteroids are scarce probably because of short treatment durations (maximum three months in PB leprosy) and an improved cellular immunity with HAART.

## Conclusion

Seventy-three cases of leprosy as IRIS in PLHIV on HAART were reported since 2003. This clinical presentation seems rare but is probably underdiagnosed and underpublished but should be known by specialists involved in management of HIV and/or leprosy. For the vast majority of these immunocompromised PLHIV, initiation of HAART reveals paucibacillary BT leprosy associated with or quickly followed by a T1R. T1R is sometimes intense with several cases of ulcerated skin lesions. The response to MDT is good, as well as to systemic corticosteroids in case of neuritis, which does not seem to be more severe than in non-IRIS cases. Isolated ulcerated lesions might heal spontaneously, allowing therapeutic abstention in some cases. Physicians need to be widely informed about the possibility of unmasking leprosy infection after HAART initiation, mainly because of the risk of irreversible nerve damage. Current recommendations are to treat these patients with MDT possibly combined with corticosteroids. Large prospective studies are needed to better assess the time course of skin damage, neuritis, response to MDT, possible relapses and the long-term effects of corticosteroids. The pathophysiology of leprosy as an IRIS is still hypothetical and requires immunological

investigations assessing cellular, cytokines and molecular players within the skin and nerves. These studies will provide a better understanding of the interactions between leprosy and HIV in context of HAART, and eventually lead to the development of specific management and monitoring guidelines in these co-infected patients.

## Supporting information

**S1 Prisma Checklist. For systematic review.**
(DOCX)

**S1 Table. Epidemiological, clinical and biological characteristics of the 73 patients diagnosed with leprosy as IRIS included in the systematic review.** Legend: F: female; M: male; MtF: male-to-female; SL: skin lesion(s); T1R: type 1 reaction state; T2R: type 2 reaction state; U: ulceration; NT: neuritis; NFI: nerve fonction impairment; Se: sensitive; Mo: motor;[1]: before MDT introduction; [2]: after MDT introduction;?: doubt about the classification; NA: missing information.
(XLSX)

## Author Contributions

**Conceptualization:** Alice Mouchard, Pierre Couppié, Chloé Bertin.

**Investigation:** Alice Mouchard, Pierre Couppié.

**Methodology:** Alice Mouchard, Pierre Couppié, Chloé Bertin.

**Resources:** Jenna Graille.

**Supervision:** Romain Blaizot, Pierre Couppié, Chloé Bertin.

**Validation:** Romain Blaizot, Pierre Couppié.

**Writing – original draft:** Alice Mouchard, Romain Blaizot, Chloé Bertin.

**Writing – review & editing:** Alice Mouchard, Romain Blaizot, Chloé Bertin.

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
