## [Decision Letter · Decision Letter 0]

3 Nov 2021

Dear Dr. Mouchard,

Thank you very much for submitting your manuscript "Leprosy as immune reconstitution inflammatory syndrome in patients living with HIV: 20 years of French Guiana experience and systematic review" for consideration at PLOS Neglected Tropical Diseases. As with all papers reviewed by the journal, your manuscript was reviewed by members of the editorial board and by several independent reviewers. In light of the reviews (below this email), we would like to invite the resubmission of a significantly-revised version that takes into account the reviewers' comments. 

We cannot make any decision about publication until we have seen the revised manuscript and your response to the reviewers' comments. Your revised manuscript is also likely to be sent to reviewers for further evaluation.

Sincerely,

Johan Van Weyenbergh

Associate Editor

Gerson Penna

Deputy Editor

Reviewer's Responses to Questions

**Key Review Criteria Required for Acceptance?**

**Methods**

-Are the objectives of the study clearly articulated with a clear testable hypothesis stated?

-Is the study design appropriate to address the stated objectives?

-Is the population clearly described and appropriate for the hypothesis being tested?

-Is the sample size sufficient to ensure adequate power to address the hypothesis being tested?

-Were correct statistical analysis used to support conclusions?

-Are there concerns about ethical or regulatory requirements being met?

Reviewer #1: The study objectives are not clearly articulated.

The population has been described.

No statistical test have been carried out just reporting of descriptive statistics

Reviewer #2: The paper is well design and adhere to the necessary guideline for the systematic review. there are some points needed to be added. Please see general comments.

**Results**

-Does the analysis presented match the analysis plan?

-Are the results clearly and completely presented?

-Are the figures (Tables, Images) of sufficient quality for clarity?

Reviewer #1: The tables are too busy and can be edited just to present relevant data.

Reviewer #2: yes, the results are interesting and well presented. Although a deeper analysis can be performed using mean or median for IRIS onset for patients with IRIS/T1R and IRIS/unreactional. Also CD44+ fold increase. Please see details below.

**Conclusions**

-Are the conclusions supported by the data presented?

-Are the limitations of analysis clearly described?

-Do the authors discuss how these data can be helpful to advance our understanding of the topic under study?

-Is public health relevance addressed?

Reviewer #1: (No Response)

Reviewer #2: COnclusions could be shortened.

**Editorial and Data Presentation Modifications?**

Reviewer #1: The manuscript by Mouchard et al, retrospectively investigates the IRIS in Leprosy in PLWHIV due to HAART. Although might be important for the medical community for that region to have the information, the manuscript itself is wanting in the way the data has been presented.

My biggest concern is after 20 years of exhaustive literature search among the n =246, only 22 had HIV co-infection and only 6 people could be found to have leprosy-IRIS? That suggests that it is not as much a healthcare problem as the authors indicate in the introduction to build their argument for the study.

The authors mention that leprosy and HIV seem to be reported to progress independently, but they have evidence that due to access to retroviral therapy, that’s not the case, which might be true but 6 individuals in 20 years doesn’t inspire confidence for that argument.

The manuscript is a combination of case-reports and systematic review, and the title needs to reflect that.

The manuscript needs to be shortened to drive home a succinct point, and as it is more of that perhaps the cases were underreported or underdiagnosed and a call for attention on that subject, the conclusions should reflect that.

Minor points:

• The spelling of French Guiana needs to be consistent

• The authors should include a line number when submitting manuscripts helps reviewers to pinpoint the changes.

• Discussion needs to be shortened a lot to include only major point of which there is a few.

Reviewer #2: (No Response)

**Summary and General Comments**

Reviewer #1: (No Response)

Reviewer #2: Leprosy as immune reconstitution inflammatory syndrome in patients living with HIV : 20 years of French Guiana experience and systematic review. The paper is important and combine clinical and laboratory data to better describe the IRIS phenomena that triggers leprosy.

There is some information that could be useful. For either the retrospective study or the systematic review, could authors retrieve whether patients are contact (social or household) that could be added. Also, if authors could have number of CD4+ on the onset of IRIS/reactional leprosy as compared to IRIS/unreactional leprosy, that could help understand the pathogenesis of IRIS and more importantly leprosy. In this regard, these groups have different time of onset for IRIS/reaction or IRIS/leprosy per se?

For a broader readership IRIS types should be briefly introduced in the methods.

Authors should consider discuss the onset of IRIS for other infectious diseases such as tuberculosis. Similarities and differences should be pointed out.

Minor issues

English can be improved.

In the last paragraph of the introduction “updated systemic review” should be updated systematic review, right?

Please include reference of Deps and Lockwood: “We searched the files of all patients followed for leprosy in the dermatology department and extracted all files of HIV infected patients meeting the criteria defined by Deps and Lockwood.”

In the table 2 the number of weeks in two patients are “16?”. What does it mean?

In the discussion the text: “IRIS is probably related to an early restoration of memory T-cell activity leading to an excessive inflammatory response in the presence of a latent infectious agent [61]. Despite the fact that (…)” needs better formatting.
---

## [Decision Letter · Decision Letter 1]

8 Feb 2022

Dear Dr. Mouchard,

We are pleased to inform you that your manuscript 'Leprosy as immune reconstitution inflammatory syndrome in patients living with HIV : description of French Guiana's cases over 20 years and systematic review of the literature' has been provisionally accepted for publication in PLOS Neglected Tropical Diseases.

Best regards,

Johan Van Weyenbergh

Associate Editor

Gerson Penna

Deputy Editor

Reviewer's Responses to Questions

**Key Review Criteria Required for Acceptance?**

**Methods**

-Are the objectives of the study clearly articulated with a clear testable hypothesis stated?

-Is the study design appropriate to address the stated objectives?

-Is the population clearly described and appropriate for the hypothesis being tested?

-Is the sample size sufficient to ensure adequate power to address the hypothesis being tested?

-Were correct statistical analysis used to support conclusions?

-Are there concerns about ethical or regulatory requirements being met?

Reviewer #1: This was a revision and the authors have successfully incorporated suggestions of both the reviewers, I went through it in detail and am happy with the current version of the manuscript.

Reviewer #2: All suggestions were accepted and changes improved the presentation of the methods for the systematic review.

**Results**

-Does the analysis presented match the analysis plan?

-Are the results clearly and completely presented?

-Are the figures (Tables, Images) of sufficient quality for clarity?

Reviewer #1: Yes

Reviewer #2: Few important improvements comparing mean tome for the outcome of IRIS among reactional and unreactional patients was introduced. Although data is based in a small sample, it is an important remark since few reports are available.

**Conclusions**

-Are the conclusions supported by the data presented?

-Are the limitations of analysis clearly described?

-Do the authors discuss how these data can be helpful to advance our understanding of the topic under study?

-Is public health relevance addressed?

Reviewer #1: Yes

Reviewer #2: Discussion and conclusion were improved.

**Editorial and Data Presentation Modifications?**

Reviewer #1: (No Response)

Reviewer #2: no need

**Summary and General Comments**

Reviewer #1: (No Response)

Reviewer #2: (No Response)

---

## [Editor Report · Acceptance letter]

20 Feb 2022

Dear Mme Mouchard,

We are delighted to inform you that your manuscript, "Leprosy as immune reconstitution inflammatory syndrome in patients living with HIV : description of French Guiana's cases over 20 years and systematic review of the literature," has been formally accepted for publication in PLOS Neglected Tropical Diseases.

Best regards,

Shaden Kamhawi

co-Editor-in-Chief

Paul Brindley

co-Editor-in-Chief
